# The DNA methylation landscape of giant viruses

Sandra Jeudy[1], Sofia Rigou [1], Jean-Marie Alempic[1], Jean-Michel Claverie [1], Chantal Abergel[1] & Matthieu Legendre [1]✉

DNA methylation is an important epigenetic mark that contributes to various regulations in all domains of life. Giant viruses are widespread dsDNA viruses with gene contents overlapping the cellular world that also encode DNA methyltransferases. Yet, virtually nothing is known about the methylation of their DNA. Here, we use single-molecule real-time sequencing to study the complete methylome of a large spectrum of giant viruses. We show that DNA methylation is widespread, affecting 2/3 of the tested families, although unevenly distributed. We also identify the corresponding viral methyltransferases and show that they are subject to intricate gene transfers between bacteria, viruses and their eukaryotic host. Most methyltransferases are conserved, functional and under purifying selection, suggesting that they increase the viruses' fitness. Some virally encoded methyltransferases are also paired with restriction endonucleases forming Restriction-Modification systems. Our data suggest that giant viruses' methyltransferases are involved in diverse forms of virus-pathogens interactions during coinfections.

---

[1] Aix Marseille Univ., CNRS, IGS, Information Génomique & Structurale (UMR7256), Institut de Microbiologie de la Méditerranée (FR 3489), Marseille, France. ✉email: legendre@igs.cnrs-mrs.fr

Methylation of DNA is an important class of epigenetic modification observed in the genomes of all domains of life. In eukaryotes, it is involved in biological processes as diverse as gene expression regulation, transposon silencing, genomic imprinting, or development[1–4]. In prokaryotes, DNA methylation often results from the targeted activity of methyltransferases (MTases) that are components of the restriction-modification (R-M) systems, which involve methylation and restriction activity. Within these systems, restriction enzymes (REases) cleave the DNA only if the shared recognized motifs are unmethylated[5]. This provides prokaryotes with a powerful weapon against foreign DNA, such as the one of infecting viruses[6]. Besides R-M systems, prokaryotic DNA MTases may occur without cognate REases, in which case they are coined orphan. Prokaryotic orphan MTases are involved in the regulation of gene expression[7,8], DNA replication[9], DNA repair[10], and cell cycle regulation[11].

Outside of the cellular world, some DNA viruses exploit DNA methylation as a mechanism to regulate their replication cycle. For instance, the transition from latent to lytic infection in Epstein–Barr Virus is mediated by the expression of genes that are silenced or transcribed according to the methylation status of their promoter[12]. Iridoviruses and ascoviruses sometimes exhibit heavily methylated genomes and encode their own MTases[12]. *Phycodnaviridae* members also encode functional MTases able to methylate their own DNA[13], and among them some chloroviruses encode complete R-M systems with their associated REases[14]. These endonucleases, packaged in the virions, contribute to the degradation of host DNA either to allow for the recycling of deoxynucleotides, or to inhibit the expression of host genes by shifting the transcription from host to viral DNA[14].

Over the last 15 years several viruses whose particles are large enough to be seen by light microscopy were discovered[15–23]. These so-called giant viruses exhibit DNA genomes as large and complex as prokaryotes[16], or even parasitic eukaryotes[22]. A growing body of metagenomics surveys shows that they are widespread on the planet in diverse environments[24,25]. The first family of giant viruses to be discovered, the *Mimiviridae*, have megabase-sized AT-rich linear genomes packaged in icosahedral capsids[16–18]. Intriguingly some *Mimiviridae* members are infected by smaller 20-kb-dsDNA viruses, dubbed virophages[26,27] and sometimes found in association with 7-kb-DNA episomes called transpovirons[28,29]. In contrast to *Mimiviridae*, the pandoraviruses have GC-rich linear genomes, twice as big with up to 2.5 Mb, packaged in amphora-shaped capsids[22,30]. Again different, pithoviruses[20] and cedratviruses[21,31] have smaller circular AT-rich genomes, ranging from 400 to 700 Kb, and packaged in the largest known amphora-shaped capsids. Thus, although these different giant viruses infect the same hosts (amoebas of the *Acanthamoeba* genus), they exhibit different morphological features, replication cycles, gene contents, and potential epigenomic modifications. To date virtually nothing is known about the epigenomes of these giant viruses. In particular, the methylation status of their DNA is unknown despite the presence of predicted MTases in their genomes. Pandoravirus dulcis for instance encodes up to five different DNA MTases (UniprotKB IDs: [S4VR68], [S4VTY0], [S4VS49], [S4VUD3], and [S4VQ82])[22]. Yet, it remains to be assessed whether any of these enzymes methylate the viral DNA.

Most of the genome-wide studies of eukaryotic DNA methylation have been performed using bisulfite sequencing techniques[32]. These approaches only detect 5-methyl-cytosine modifications and are thus not well suited for the analysis of prokaryotic-like epigenomic modifications, mostly composed of $N^6$-methyl-adenines and $N^4$-methyl-cytosines. However, the recently developed single-molecule real-time (SMRT) sequencing method overcomes this limitation[33]. Briefly this approach analyses the kinetics of incorporation of modified nucleotides by the polymerase compared to the non-modified ones. The Inter-Pulse Duration ratio (IPDr) metric can then be computed for each genomic position, and makes it possible to map all modified nucleotides and methylated motifs along the genome. This approach is now extensively used to study the methylation landscapes of isolated bacteria[34], archaea[35], and even prokaryotic metagenomes[36].

Here, we use SMRT sequencing to survey the complete methylome of a large spectrum of giant viruses. We analyze two distinct *Mimiviridae* members and their associated transpovirons, as well as a virophage[28]. We also survey a *Marseilleviridae* member[37], five distinct pandoraviruses[22,30,38], a mollivirus[23] and a pithovirus[20]. Finally, we isolate a new cedratvirus (cedratvirus kamchatka) that we sequence using SMRT sequencing to assess its methylome. Furthermore, we thoroughly annotate MTases and REases contained in all these genomes and analyze their phylogenetic histories. Our findings reveal that DNA methylation is widespread among giant viruses and open new avenues of research on its role in their population dynamics.

## Results

**Methylome and MTase gene contents of giant viruses.** We gathered PacBio SMRT data of diverse families from previously published genomes sequenced by our group to analyze the DNA methylation profile of a wide range of giant viruses. SMRT genomic data were collected for the following viruses: the *Mimiviridae* member moumouvirus australiensis and its associated transpoviron[28], the *Marseilleviridae* member melbournevirus[37] and five pandoraviruses (pandoravirus celtis[38], pandoravirus dulcis[30], pandoravirus neocaledonia[30], pandoravirus quercus[30], and pandoravirus salinus[30]). In addition, we resequenced on the PacBio platform the complete genomes of mollivirus sibericum, pithovirus sibericum, the *Lavidaviridae* member zamilon vitis and the megavirus vitis *Mimiviridae* member together with its associated transpoviron. Finally, we sequenced a newly isolated strain of cedratvirus (cedratvirus kamchatka). The datasets are listed in Supplementary Table 1. The sequence data obtained from the whole collection corresponds to an average coverage of 192-fold.

We then aligned the SMRT reads to their corresponding reference genomes (see Supplementary Table 1) and computed IPDr at each genomic position (see Methods). These genome-wide profiles were used to identify overrepresented sequence motifs at positions with high IPDr values. All the identified motifs were palindromic and prone to either $N^4$-methyl-cytosine or $N^6$-methyl-adenine methylations (Fig. 1). As a control, we applied the same procedure to DNA samples of mollivirus sibericum and pithovirus sibericum subjected to whole genome amplifications (WGA), which in principle erase methylation marks[33]. As expected, no overrepresented methylated motif was detected in these controls, and the median IPDr of the motifs previously detected in the wild-type datasets were basal here (see Supplementary Fig. 1 and Fig. 1).

In parallel, we analyzed the DNA MTases encoded in these genomes and predicted their target sequences based on their homology with characterized MTases (see Methods). All the MTases for which a target site could be predicted were putative type II MTases. In 15 out of the 19 cases (79%) where a MTase target could be predicted or a methylated motif detected, we found an agreement between the two (Fig. 1). It is worth mentioning that this result also highlights the reliability of MTases' targets predictions based on protein homology.

***Marseilleviridae* members encode complete R-M systems.** Melbournevirus encodes a DNA MTase (mel_016) predicted to

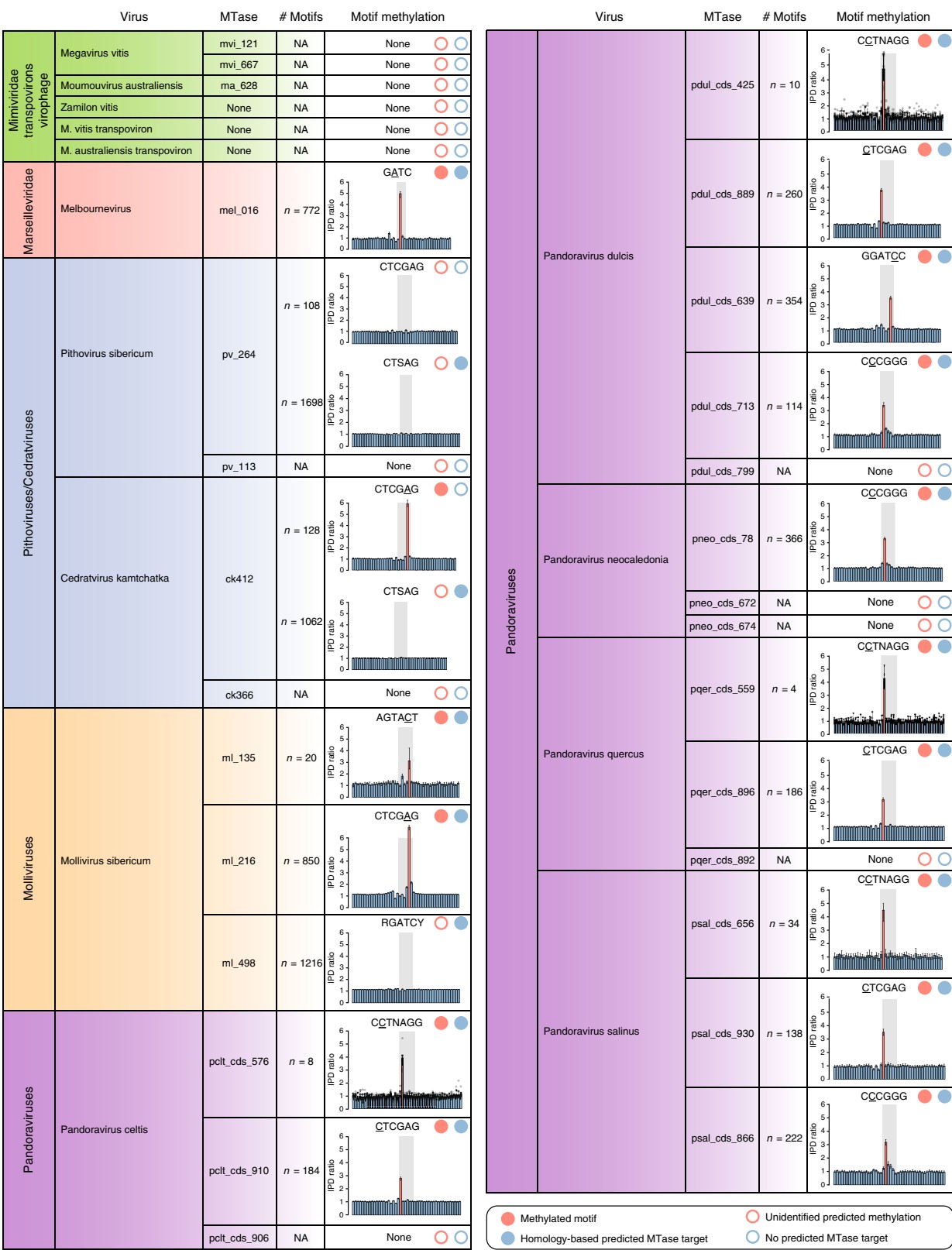

**Fig. 1 Encoded MTases and targeted methylated motifs in the giant viruses' genomes.** The encoded DNA MTases of each virus are shown along with the number of occurrences of the predicted targets (if any) on both strands of the cognate genomic sequence. Modified nucleotides within the motifs are underlined. Red circles indicate methylated motifs experimentally verified from SMRT data (filled circles) or with unidentified predicted methylation (empty circles). Likewise, predicted (filled circle) and not predicted (empty circle) targets based on sequence homology of the encoded MTase are shown using blue circles. Bar graphs correspond to the median IPDr profiles of the motifs (gray region) and the surrounding 20 nucleotides on each side. Each bar displays the median IPDr value and a 95% confidence interval (error bars) based on 1000 bootstraps. These statistics are derived from the number of occurrences ($n$) of the motifs in each genome. Red bars correspond to positions with significantly high IPDr values. Individual data points are displayed for viruses with $n \leq 10$.

target GATC sites. Our data confirm that GATC motifs were modified (underlined characters indicate methylated bases) with N[6]-methyl-adenines (Fig. 1). We then searched for the possible cognate REase in the genomic vicinity of mel_016 and identified the neighboring mel_015 gene as a candidate. Although the encoded protein does not exhibit a recognizable motif using standard domain search tools[39,40], a search against REbase, the database dedicated to R-M systems[41], identified it as a probable GATC-targeting REase. Moreover, the mel_015 protein is similar (blast $E$-value $= 2 \times 10^{-33}$) to the *Paramecium bursaria* Chlorella virus 1 (PBCV-1) CviAI REase known to target GATC sites. Melbournevirus thus encodes a complete R-M system.

The N[6]-methyl-adenine modification is typical of prokaryotic MTases. Since melbournevirus is a eukaryotic virus, our finding immediately questioned the evolutionary history of its encoded R-M system. We thus reconstructed the phylogeny of the complete system, including the MTase and the REase (see Methods). The mel_016 MTase strikingly branches within the prokaryotes along with other viruses, mostly chloroviruses and members of the *Marseilleviridae* and *Mimiviridae* of the proposed mesomimivirinae subfamily[42] (Supplementary Fig. 2A). In agreement with its enzymatic activity, this phylogeny suggests that the melbournevirus MTase was acquired from a prokaryote. Likewise, the phylogeny of the mel_015 REase suggests its prokaryotic origin (Supplementary Fig. 2B). Altogether, these results support a relatively ancient acquisition from prokaryotes as the origin of the complete marseilleviruses R-M system.

Surprisingly, we did not identify orthologues of the melbournevirus R-M system in all *Marseilleviridae* members. As shown in Fig. 2, only 5 out of the 13 marseilleviruses genomes contain both a MTase and a REase, always encoded next to each other. The other marseilleviruses encode neither the MTase nor the REase. The *Marseilleviridae* phylogeny based on core genes (see Methods) clearly coincides with its dichotomous distribution (Fig. 2). All the clade A members of the *Marseilleviridae* encode a complete R-M system, while the others do not. This suggests that the marseilleviruses R-M system was acquired by the clade A ancestor. It is worth noticing that once acquired, the R-M system was maintained, with none of the two enzymes undergoing pseudogenization. This suggests that the encoded R-M system has a functional role in these viruses.

### Activity of the *Marseilleviridae* members R-M system on non-self DNA. Once methylated by the mel_016-encoded enzyme, we expect the viral DNA to be protected from its digestion by the

cognate mel_015 REase. If not, the melbournevirus genome would be theoretically fragmented into 387 fragments of 954 nt on average. We verified this prediction by conducting DNA restriction experiments using two endonucleases targeting GATC sites: DpnI and DpnII. The former only cleaves DNA at modified GATC sites containing a N[6]-methyl-adenine, while the latter conversely only cleaves unmethylated GATC sites. Figure 3 demonstrates that melbournevirus DNA is digested by DpnI but not by DpnII. Thus, assuming that mel_015 REase is functional, we can infer that melbournevirus is able to protect its own genome from its encoded R-M system digestion. As a control, we reproduced the above experiment with the DNA of noumeavirus[43], a *Marseilleviridae* member belonging to the clade B that do not encode a R-M system (Fig. 2). As expected, the noumeavirus genome is digested by DpnII but not by DpnI (Fig. 3). This demonstrates that noumeavirus DNA is not methylated at GATC sites and is thus susceptible to degradation by a co-infecting marseillevirus bearing a functional R-M system.

To verify whether the *Acanthamoeba castellanii* host genome was sensitive to DNA degradation at GATC sites we performed digestion assays using the same enzymes. According to its sequence (GenBank accession AEYA00000000), the *A. castellanii* genome should be fragmented into 171,298 pieces of 273 nt on average. Surprisingly, the *A. castellanii* genome is cleaved by both enzymes (Fig. 3). This indicates that the host DNA contains a mixture of methylated and unmethylated GATC sites. However, the restriction profiles show that unmethylated positions are in larger proportion than methylated ones.

Since the host DNA is (at least partially) unprotected from the marseilleviruses encoded R-M systems we next assessed its potential degradation during a melbournevirus infection. As shown in Supplementary Fig. 3, A. castellanii DNA is not degraded during the infection. As expected, the infection of *A. castellanii* with noumeavirus, which do not encode the GATC R-M system, do not alter its DNA. Hence, marseilleviruses encoded R-M systems do not contribute to host DNA degradation.

To further investigate the role of the marseilleviruses R-M systems we analyzed the timing of expression of both enzymes (the MTase and the REase) during a melbournevirus infection. Figure 4 shows that the mel_015 REase gene is first transcribed between 30 min and 45 min post infection, followed by the mel_016 MTase gene expressed between 45 min and 1 h post infection. In addition, proteomic data of the melbournevirus virion from[43] confirm that the mel_016 MTase protein is packaged in the particle whereas the REase is not.

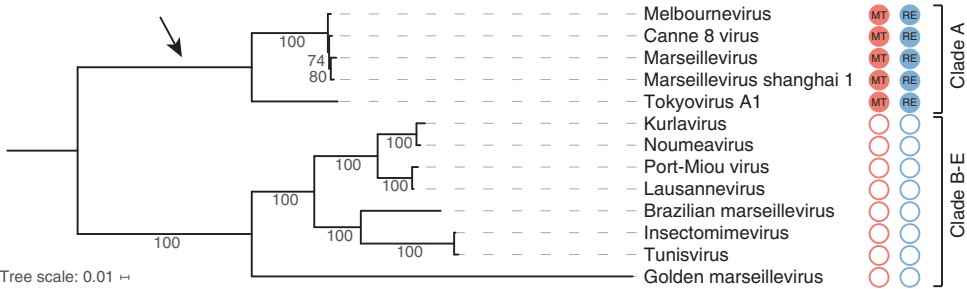

**Fig. 2 Presence/absence of R-M systems in the *Marseilleviridae* family.** Phylogeny of the *Marseilleviridae* completely sequenced viruses with the following GenBank accessions: melbournevirus (KM275475.1)[37], cannes 8 virus (KF261120.1)[88], marseillevirus (GU071086.1)[89], marseillevirus shanghai 1 (MG827395.1), tokyovirus A1 (AP017398.1)[90], kurlavirus (KY073338.1)[91], noumeavirus (KX066233.1)[43], port-Miou virus (KT428292.1)[92], lausannevirus (HQ113105.1)[48], brazilian marseillevirus (KT752522.1)[93], insectomimevirus (KF527888.1)[94], tunisvirus (KF483846.1)[95] and golden marseillevirus (KT835053.1)[96]. The phylogeny was based on protein sequence alignments of the 115 strictly conserved single copy orthologues (see Supplementary Data 1). The tree was calculated using the best model of each partitioned alignment as determined by IQtree[82]. Bootstrap values were computed using the UFBoot[84] method. The red and blue filled circles highlight the presence of encoded marseilleviruses R-M systems MTases and REases respectively. The empty circles highlight the absence of the MTase (in red) and the REase (in blue). The arrow points to the most parsimonious acquisition of a R-M system in the *Marseilleviridae* family.

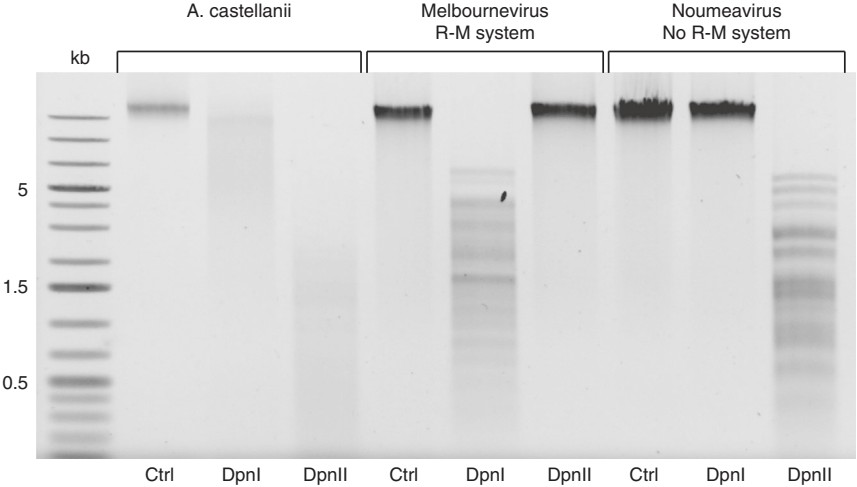

**Fig. 3 Host and marseilleviruses DNA protection against GATC-targeting REases.** Agarose gel electrophoresis analysis of *A. castellanii*, melbournevirus and noumeavirus DNA digested with GATC-targeting restriction enzymes. Restriction patterns using DpnI and DpnII enzymes are presented with control DNA. DpnI cleaves DNA at G<u>A</u>TC sites containing N[6]-methyl-adenines and DpnII at GATC sites containing unmethylated adenines. This experiment was repeated twice with similar results.

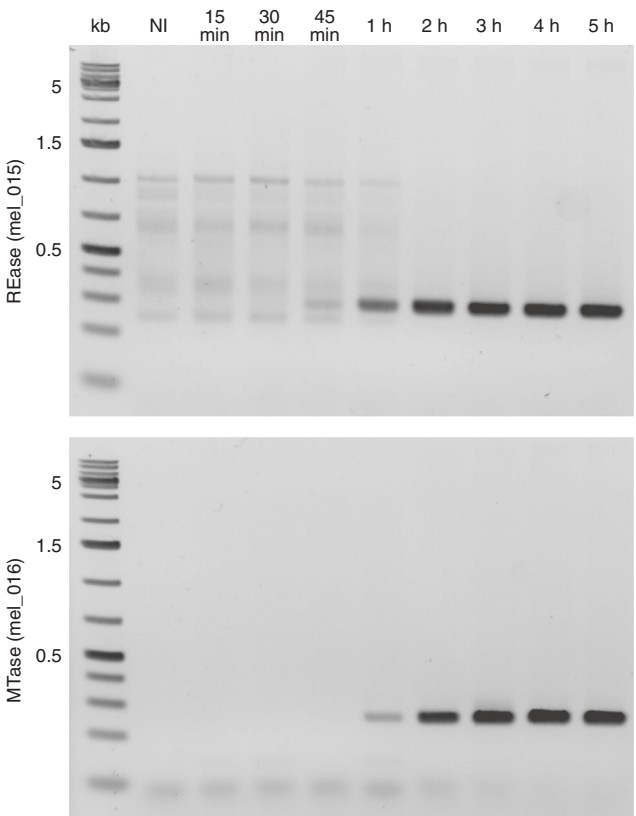

**Fig. 4 Expression timing of the melbournevirus R-M system MTase and REase.** Shown are the RT-PCRs of the transcripts corresponding to the mel_015 and mel_016 genes during a melbournevirus infection. Times (post infection) are listed on the top of the figure. NI corresponds to non-infected. This experiment was repeated twice with similar results.

**The newly isolated cedratvirus kamchatka.** Cedratviruses are giant viruses morphologically and to some extant genetically related to pithoviruses[31]. Four completely sequenced genomes are available today: cedratvirus A11[21], cedratvirus zaza[44], cedratvirus lausannensis[31], and brazilian cedratvirus[44]. We isolated a new strain of cedratvirus (named cedratvirus kamchatka) from a muddy grit soil sample collected near a lake at kizimen volcano, Kamchatka (Russian Federation N 55°05'50 E 160°20'58) (see Methods). SMRT sequencing was used to characterize both its genome and methylome. The cedratvirus kamchatka genome was assembled into a circular 466,767 bp DNA molecule (41% G + C), predicted to encode 545 protein-coding genes The genome size and topology was confirmed by pulsed field gel electrophoresis (PFGE) (Supplementary Fig. 4A).

The phylogenetic tree computed from the pithoviruses and cedratviruses core genes shows that they cluster in well-separated groups (Supplementary Fig. 4B). Their orthologous proteins share an average of 46% identical residues. The available cedratviruses appear to split into three distinct clades: clade A contains cedratvirus A11, cedratvirus zaza and cedratvirus lausannensis, clade B contains cedratvirus kamchatka, and clade C contains brazilian cedratvirus (Supplementary Fig. 4B). This classification might be challenged as new strains will be characterized. We found that 51 of the 545 cedratvirus kamchatka genes were unique to this strain compared to the other cedratviruses. According to the presence/absence of pseudogenes in the other strains, as detected using tblastn, we designed a putative evolutionary scenario for each of these genes (see Supplementary Table 2). As previously discussed for pandoraviruses[30,38], the process of de novo gene creation seems to participate to the shaping of cedratviruses genomes. Interestingly, among the 51 genes unique to cedratvirus kamchatka only one has a clear predicted function: the ck412 DNA MTase.

**DNA methylation is widespread but unevenly distributed among giant viruses genomes.** The SMRT sequencing of cedratvirus kamchatka and the other methylome datasets show that the various giant virus families exhibit distinct methylation features. Whereas the genomes of pithoviruses and *Mimiviridae* members are devoid of DNA modifications, those of molliviruses, pandoraviruses and cedratviruses clearly contain methylated nucleotides (Fig. 1).

More specifically, cedratvirus kamchatka DNA is methylated at CTCGAG motifs (Fig. 1). Although the ck412 DNA MTase has a slightly different predicted target (CTSAG), it is probably responsible for the CTCGAG methylation as CTSAG motifs are

not methylated (Fig. 1). Cedratvirus kamchatka ck366 gene encodes an additional predicted FkbM domain-containing MTase for which the predicted specific target, if any, is unknown. Importantly, we found no REase associated with the cedratvirus kamchatka predicted MTases.

Pithovirus sibericum encodes two DNA MTases: pv_264 predicted to target the CTSAG motif and pv_113, an FkbM domain-containing MTase targeting an unknown site. Yet, pithovirus sibericum DNA exhibit no methylated sites (including CTSAG and CTCGAG) (Fig. 1). Surprisingly, the RNA-seq data from[20] shows that both transcripts are significantly expressed all along the replication cycle (see Supplementary Tables 3 and 4). Finally, none of the genes surrounding the MTases are predicted to encode a functional REase. Thus, according to our SMRT-seq data pithovirus sibericum MTase-like proteins do not methylate the viral DNA even though they are expressed.

Although megavirus vitis encodes a type 11 domain MTase (mvi_121) and an FkbM domain-containing MTase (mvi_667) shared with moumouvirus australiensis (ma_628), we cannot infer their DNA target specificities from sequence homology or experimental evidences since none of the two genomes appear to be methylated (Fig. 1). In addition to these MTase-like candidates, megavirus vitis and moumouvirus australiensis encode a 6-O-methylguanine-DNA methyltransferase (mvi_228/ma_196) probably involved in DNA repair. We also surveyed the DNA methylation of the *Mimiviridae* members' mobilome, namely the zamilon vitis virophage, the megavirus vitis transpoviron and the moumouvirus australiensis transpoviron. These genomes that do not encode DNA MTases are not methylated (Fig. 1). Collectively this data show that the putative DNA MTases encoded by the *Mimiviridae* members infecting *Acanthamoeba* do not methylate the viral or mobilome DNA.

In contrast, all the surveyed pandoraviruses' genomes are methylated (Fig. 1). Unexpectedly they exhibit $N^4$-methyl-cytosines instead of the $N^6$-methyl-adenines found in *Marseilleviridae* members and cedratviruses. The number of distinct methylated motifs is also quite variable: a single one in pandoravirus neocaledonia, but two in pandoravirus celtis and pandoravirus quercus, three in pandoravirus salinus, and up to four in pandoravirus dulcis. We successfully assigned each of all methylated motifs to their cognate encoded MTases. However, none appeared to be associated with a REase. In addition, we found a MTase type 25 domain-containing gene in pandoravirus neocaledonia (pneo_cds_672), as well as a MTase type 11 domain gene (pneo_cds_674) with orthologs in pandoravirus dulcis, pandoravirus quercus, and pandoravirus celtis (pdul_cds_799, pqer_cds_892 and pclt_cds_906). None of them had predicted DNA targets.

Finally, the mollivirus sibericum genome is prone to both types of modification: $N^4$-methyl-cytosines and $N^6$-methyl-adenines. It is methylated at the AGCACT sites by the ml_135 encoded MTase and at the CTCGAG sites by the ml_216 MTase (Fig. 1). A third MTase encoded by this genome (ml_498) is predicted to recognize the RGATCY sites but our methylome data clearly show that they are not methylated (Fig. 1). We first suspected that the ml_498 gene was not transcribed but transcriptomic data from[23] clearly show that ml_498 is expressed, mostly in the early phase of the infection (see Supplementary Table 5). However, the analysis of the gene structure shows a long 5'UTR, suggesting a N-terminal truncation of the protein (Supplementary Fig. 5A) compared to its homologs (Supplementary Fig. 5B). As a single frameshift is sufficient to restore the N-terminal part of the protein, ml_498 probably underwent a recent pseudogenization.

The activity of ml_216 was further confirmed by restriction experiments showing that mollivirus sibericum DNA is cleaved after WGA at CTCGAG sites but not in wild-type conditions

(Supplementary Fig. 6A). By contrast, the RGATCY sites are not protected, as expected from the ml_498 loss of function. Interestingly while the RGATCY, AGTACT and CTCGAG sites are unmethylated in host DNA, the CCCGGG motifs are protected against degradation (Supplementary Fig. 6A, B). This probably corresponds to CpG methylation of the host DNA.

None of the mollivirus sibericum MTases appear to be associated with a corresponding REase. This lack of nuclease activity was confirmed by the absence of host DNA degradation during the infection cycle (Supplementary Fig. 7). One might expect that the host DNA is protected against putative CTCGAG and AGTACT targeting viral REases by endogenously encoded MTases. We exclude this possibility since host DNA is sensitive to degradation at those sites in uninfected conditions (Supplementary Fig. 6B).

**The complex evolutionary history of giant viruses' MTases.** In the *Marseilleviridae* family, we observed methylation patterns typical of prokaryotic MTases. This raised the question of their evolutionary histories. In Fig. 5 we now present a global phylogenetic analysis of all giant viruses' MTases analyzed in this work (Fig. 1). They clearly do not share a common origin, and appear partitioned in five main groups, interspersed among bacterial and occasional amoebal homologs.

First, one observes that most viral MTases are either embedded within clusters of prokaryotic sequences (green, and red groups in Fig. 5) or constitute sister groups of prokaryotes (orange, purple and blue groups). Thus, as for *Marseilleviridae* members these MTases are most likely of bacterial origin.

Of special interest, the tree also exhibits MTases encoded by different *Acanthamoeba* species (see GenBank accessions in Fig. 5), the main known hosts of giant viruses. For instance, the closest ml_135 mollivirus MTase homolog is found in *A. polyphaga*, suggesting a recent exchange between virus and host. The direction of the gene transfer cannot be determined from these data. However, in the red and orange groups (Fig. 5) other acanthameoba homologs appear well nested within pandoraviruses MTases. This supports transfers occurring from the giant viruses to the host genome. We also noticed divergent MTases attributed to various *Acanthamoeba* species in the purple group. However, a closer inspection of the taxonomic assignment of the corresponding contigs indicates that they are bacterial sequences, probably from the *Bradyrhizobiaceae* family (see Supplementary Fig. 8). These bacteria are amoeba resistant intracellular microorganisms[45] that probably contaminated the eukaryotic host sequencing project.

The purple group also contains three orthologous MTases targeting CTCGAG sites: ml_216 from mollivirus sibericum, pv_264 from pithovirus sibericum and ck412 from cedratvirus kamchatka. These viruses belong to two distinct viral families but infect similar hosts. It is thus likely that these MTases were recently exchanged between these viruses. In the green group there are also three orthologous MTases from distinct viral families: pdul_cds_639 and ppam_cds_578 from pandoravirus dulcis and pandoravirus pampulha respectively, as well as ml_498 from mollivirus sibericum. The two pandoraviruses are closely related with an average of 83% sequence identity between shared proteins. As this MTase is not found in other pandoraviruses (Supplementary Fig. 9), a gene exchange might have occurred between the pandoravirus pampulha-pandoravirus dulcis ancestor and mollivirus sibericum.

**Selection pressure acting on giant viruses' MTases.** Following the above phylogenetic analysis of the giant viruses MTases, we investigated the selection pressure acting on them. We first

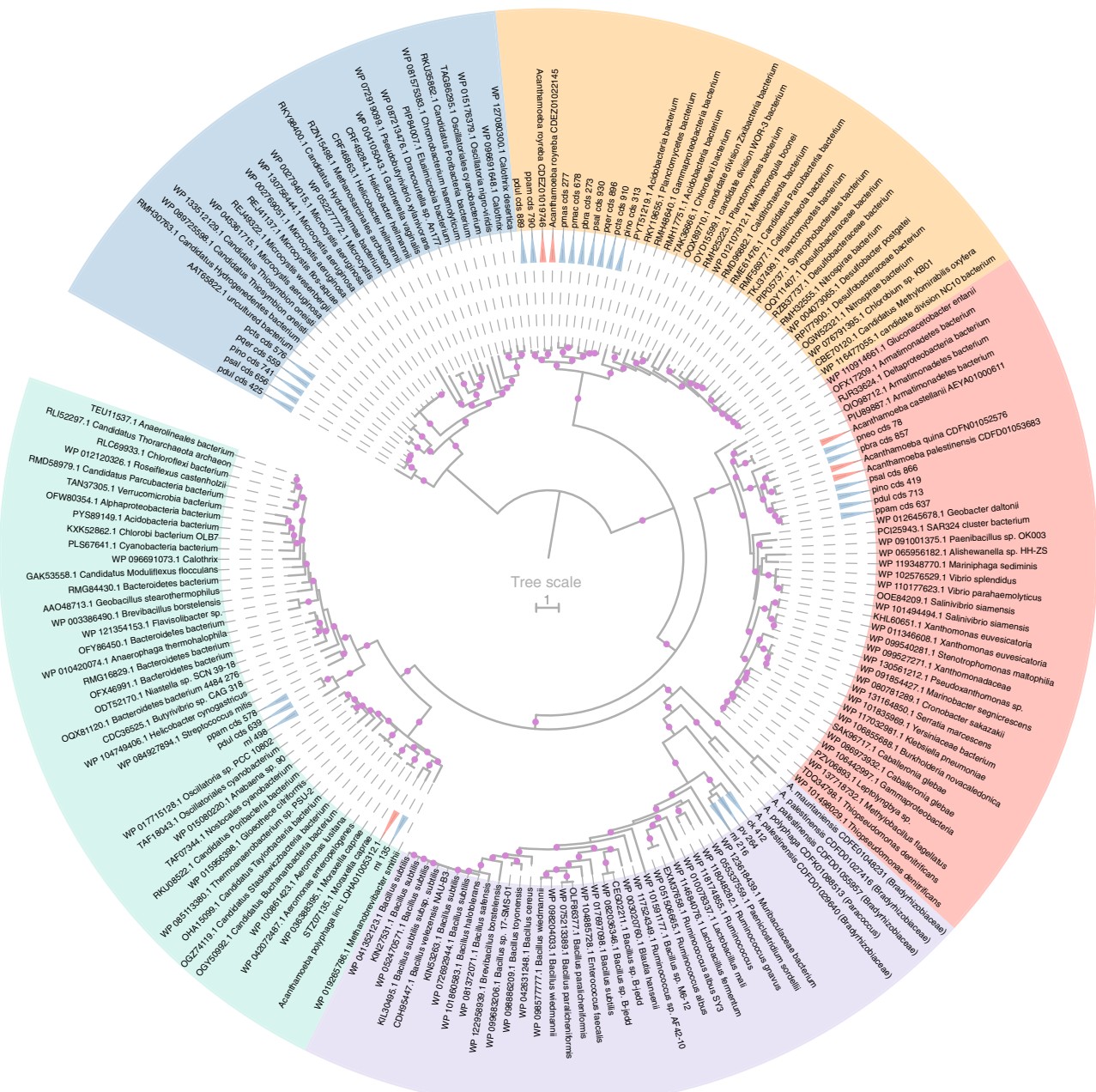

**Fig. 5 Phylogenetic tree of the giant viruses MTases.** Phylogenetic tree of the giant viruses' MTases along with prokaryotic and eukaryotic homologs. The blue triangles mark viral genes, the red ones eukaryotic genes and the unmarked genes are prokaryotic. The tree was computed using the LG + R6 model from a multiple alignment of 678 informative sites. Bootstrap values were computed using the UFBoot[84] method from IQtree[82]. All branches with support value > 80 are highlighted using purple circles. The GenBank accessions and taxonomic assignations extracted from GenBank entries are shown. The tree was rooted using the midpoint rooting method. The tree was split into five subgroups highlighted using different colors (blue, orange, red, purple, and green).

noticed that some MTases were conserved for long periods of time in various viral families. The CTCGAG and CCCGGG targeting MTases, most likely gained by a Pandoravirus ancestor, remain present in most of the extant Pandoraviruses (Supplementary Fig. 9). Likewise, the marseilleviruses R-M MTase and the CCTNAGG pandoravirus targeting MTase were kept in almost all members of their respective clades (Fig. 2 and Supplementary Fig. 9). By contrast, the ml_498 mollivirus sibericum MTase was found to be recently pseudogenized (Supplementary Fig. 5). In addition, the pino_cds_419 gene from pandoravirus inopinatum and the ppam_cds_578 from pandoravirus pampulha

are most likely truncated pseudogenes, even though we do not have SMRT data to confirm their loss of function.

We then computed the ratios (ω) of non-synonymous (dN) to synonymous (dS) substitution rates to quantify the selection pressure acting on the MTases. The ω of MTases with predicted targets were calculated using Codeml[46] according to three different models (see Methods). We then selected the best fitted models using likelihood ratio tests (LRT) to determine whether ω were significantly different from one. As shown in Supplementary Table 6, the majority (11/20) of the MTases had a ω significantly smaller than one and the rest could not be statistically

distinguished from neutral evolution. This indicates that most giant viruses MTases are under purifying selection.

## Discussion

Following the initial description of mimivirus[16], the last decade has seen an acceleration in the pace of discovery of giant viruses, now distributed in multiple different families[15,18,20–23,47], both thanks to the physical isolation of new specimens and to the rapid accumulation of metagenomics data[25]. Although the number of genomic sequences steadily increased during this period, the epigenomic status of giant viruses remained virtually unknown. Yet, the presence of numerous predicted DNA modification functions in their gene contents, as well as histone homologs in some of them[48], suggest that epigenetic may have a general impact on giant viruses's fitness, most likely through virus-virus and host-virus interactions. Here, we presented the first investigation of the DNA methylome of a large diversity of giant viruses using SMRT sequencing. Our analyses reveal that DNA methylation is widespread as it was detected in four of the 6 distinct giant viruses' families tested (cedratviruses, molliviruses, pandoraviruses and marseilleviruses). The recent advances in SMRT sequencing of metagenomes[36] will probably soon enable the survey of cultivation-independent giant viruses[25,49] and provide further evidence to test this hypothesis.

Our detailed investigation of the DNA MTase gene contents first confirmed the ubiquity of DNA methylation in giant viruses. We identified homologs of these enzymes in all analyzed viruses, with the exception of the *Mimiviridae* members' mobilome. Although widely present in giant viruses, the number of encoded MTases (and targeted sites) is unevenly distributed. It ranges from a single one in melbournevirus and moumouvirus australiensis, up to five in pandoravirus dulcis. Even within the same family, such as the pandoraviruses, the number of encoded DNA MTases is variable. Furthermore, the number of occurrences of each methylated sites per genome is highly variable, from 4 (CCTNAGG in pandoravirus quercus) to 850 (CTCGAG in mollivirus sibericum) (Fig. 1). The range of relative frequency of these sites is even larger from $10^{-6}$ for CCTNAGG in pandoravirus quercus up to $10^{-3}$ for the GATC motif in melbournevirus. Therefore, as already noticed in prokaryotes[34–36], DNA methylation is widespread but has a patchy distribution in giant viruses.

The non-uniformity of DNA methylation in giant viruses is partially explained by the loss-of-function of some encoded MTases. For instance, the mollivirus sibericum ml_498 MTase lacks a conserved (D/N/S)PP(Y/F) motif, involved in the formation of a hydrophobic pocket that binds the targeted nucleotide[50]. This was probably caused by a recent frameshift mutation in the 5′ region of the gene. Another case is the pithovirus sibericum pv_264 MTase also unable to methylate viral DNA, although the protein does not seem to be truncated. Here, uncharacterized mutations in a critical part of the protein might be at play and explain the loss of function. Alternatively, although less likely, the lack of methylation could be the result of transient methylation where methylation marks of viral DNA are eventually erased by an unknown $N^6$-methyl adenine demethylase. Finally, one cannot exclude that SMRT-seq is not sensitive enough to detect cryptic methylation of pithovirus DNA.

As expected from their enzymatic specificities, giant viruses' MTases are all of bacterial origins (Fig. 5). More surprisingly, we found that some of them were transferred from giant viruses (mostly pandoraviruses) to *Acanthamoeba* genomes. It remains to be determined whether this is an evolutionary dead end or if the transferred enzymes are still active in the host. Although host-to-virus gene exchanges are traditionally deemed more frequent than virus-to-host transfers[51], we previously noticed that this might

not be true in pandoraviruses[30]. The picture is even more complex concerning MTases transferred between viruses, as illustrated by the orthologous MTases identified in cedratvirus kamchatka and mollivirus sibericum, two viruses from different families. If we previously noticed that some genes might be swapped between strains of pandoraviruses[38], the present case involves an exchange between viruses from totally different families only sharing the *Acanthamoeba* host. In the recently discovered mollivirus kamchatka[52], a MTase without ortholog in Mollivirus sibericum was probably acquired from a pandoravirus. In prokaryotes, the analysis of the co-occurrence of R-M systems and genetic fluxes between bacteria revealed that genetic exchanges are favored between genomes that share the same R-M systems, regardless of their evolutionary distance[53]. A similar phenomenon might be at play between molliviruses and pandoraviruses, and partially explains their shared gene content[52]. Previous analyses have also suggested that amoeba act as a genetic melting pot between intracellular bacteria[54], a concept that should now be extended to include amoeba-infecting viruses.

Our global survey of the DNA methylome of giant viruses revealed unexpected features. Besides the $N^6$-methyl-adenine modifications identified in the melbournevirus, cedratvirus kamchatka and mollivirus sibericum, we unveiled unexpected $N^4$-methyl-cytosines in the genomes of mollivirus sibericum and all tested pandoraviruses. To our knowledge, these are the first of such modifications reported for eukaryotic viruses. Some chloroviruses contain large amounts of $N^6$-methyl-adenines and 5-methyl-cytosines[55] also found in other eukaryotic viruses such as herpesviruses[12,56], and members of the *Iridoviridae*[12,57] and *Adenoviridae*[12,58] families. However, $N^4$-methyl-cytosine modifications were until now thought to be restricted to the prokaryotes and their viruses.

Another unexpected finding is the discovery of different types of modifications of the same CTCGAG site in giant viruses. If cedratvirus kamchatka and mollivirus sibericum exhibit CTCGAG motifs (with $N^6$-methyl-adenines), the pandoraviruses exhibit CTCGAG with $N^4$-methyl-cytosines (Fig. 1). The corresponding MTases belong to two distinct phylogenetic groups (green and orange groups in Fig. 5) and were acquired from distinct prokaryotes. Structural studies will be needed to elucidate how these MTases differently methylate the same DNA motif.

It was recently discovered that the AGCT tetramer is specifically eliminated from the pandoraviruses' genomes, in particular the ones belonging to the A-clade[59]. The evolutionary mechanism causing the elimination of this motif is still mysterious. One of the rejected hypothesis was that a R-M system targeting AGCT sites could be involved[59]. Our methylome data consistently confirm that the AGCT motif is not methylated in the pandoraviruses of neither clade A nor B (Supplementary Fig. 10).

Our digestion experiments revealed the presence of $N^6$-methyl-adenine-modified GATC sites in the genome of *A. castellanii* (Fig. 3). $N^6$-methyl-adenines were long thought to be restricted to prokaryotes, until several studies showed that some eukaryotes are also subject to these modifications[60,61]. In Chlamydomonas for instance, the $N^6$-methyl-adenines preferentially localize at the vicinity of transcription start sites, in the nucleosome free regions, to mark transcriptionally active genes[60]. Since the *A. castellanii* genome contains a mixture of methylated and unmethylated GATC sites we expect that similar biased modification patterns could be revealed by SMRT sequencing.

Most of the MTases (16 over 18) that had a testable (i.e., predicted) target site were found to be functional (Fig. 1). This either suggests that they were recently acquired, or that they were conserved because they increased the recipient viruses' fitness. Several evidences favor the second hypothesis. First, we found several of them conserved in entire clades (Fig. 2 and

Supplementary Fig. 9), indicating that they were retained throughout the family's radiation. Secondly, most of them are under purifying selection (Supplementary Table 6).

We observed that the complete R-M systems found in the *Marseilleviridae* members were phylogenetically related to that of the chloroviruses, in which it functions as a host DNA recycling mechanism[14]. There was thus a possibility that the marseilleviruses R-M system could play a similar role. However, our data clearly refute this hypothesis. Even though we found that the host DNA remains vulnerable at the GATC site targeted by the marseilleviruses REases (Fig. 3), the actual infection did not induce its degradation (Supplementary Fig. 3). This result is coherent with what we know about the replication cycle of these viruses[43,48]. Even if melbournevirus temporarily requires nucleus functions to initiate its replication cycle[43], most of it then proceeds in the cytoplasm, without contact with the host DNA[43]. This supports the non-involvement of marseilleviruses R-M systems, as well as other encoded REases, in the recycling of host DNA.

By contrast, our results revealed that REases corresponding to the marseilleviruses R-M system could degrade the DNA of other marseilleviruses devoid of the same system. As previously proposed for chloroviruses[14,62] this suggests that the marseilleviruses R-M systems are involved in the exclusion of other viruses in cases of multiple infections. Indeed, *Acanthamoeba* can be infected by a wide variety of giant viruses[15,63]. In this context, a R-M system becomes an efficient way for a virus to fight against competitors and increase its fitness. In addition, *Acanthamoeba* feed on bacteria and are the reservoir of many intracellular bacteria, some of them remain as cytoplasmic endosymbionts[45,64]. The marseilleviruses R-M systems could thus be involved in recycling the DNA of these intracellular parasites.

In line with such putative role in pathogen exclusion, we found a congruent pattern of expression of the melbournevirus R-M system REase and MTase. The REase is first transcribed in the early phase of the infection cycle, where it could degrade the DNA of eventual co-infecting bacteria or viruses. The MTase is then transcribed 15 min later and the enzyme finally packaged in the virion, where it could protect the viral DNA from the REase, pending the next infection.

Besides *Marseilleviridae* family members, giant viruses for which we identified MTases and observed DNA methylation do not seem to encode cognate REases. Such so-called orphan MTases are common in bacteria where they regulate various biological processes, such as replication initiation, mismatch repair or gene expression[7–11]. Accordingly, the targeted genomic positions are not uniformly distributed, with hotspots and coldspots of fully-, hemi- and unmethylated sites. Likewise, DNA modifications in phages have epigenetic roles beyond R-M systems. For instance the P1 phage Dmt-encoded MTase is involved in the control of DNA concatemers cleavage at the initiation of DNA packaging[65–67]. Again the methylated sites are clustered in the so-called pac regions. One could hypothesize a similar replication-related epigenetic role of orphan giant viruses MTases. However, giant viruses' genomes exhibit unimodal distributions of IPDr values and the corresponding motifs are globally uniformly distributed (Supplementary Fig. 11). In the case of bacterial orphan MTases involved in gene regulations the methylated sites tend to be located in the upstream non-coding regions of the regulated genes[34]. This is not true for giant viruses where methylated motifs are not enriched in intergenic regions (empirical two sided $p$-values > 0.1, see Methods) and thus not favoring such an epigenetic role.

How could we then interpret the presence of orphan MTases in giant viruses? Their role could be to protect the viral genome from its digestion by cognate REases from other *Acanthamoeba*-infecting bacteria or viruses harboring complete R-M systems. This would explain the tendency for some viruses, such as pandoravirus dulcis, to accumulate functional MTases in their genomes. In absence of the corresponding REases in the environment, the selection pressure would be relaxed on less solicited MTases, leading to their pseudogenization (Supplementary Table 6).

Giant viruses of the *Mimiviridae* family are involved in complex networks of interactions with the cellular host, the virophages, and the transpovirons[28,29,68]. It has been proposed that R-M systems could be used as an anti-virophage agent in these multipartite systems[69]. Accordingly, we investigated the role of DNA methylation in these cross talks. Our analysis of *Acanthamoeba*-infecting *Mimiviridae* members do not currently support this view, as DNA methylation does not seem to be a key player in this network (Fig. 1). However, future studies might reveal DNA methylation of the *Mimiviridae*-virophage system using different methylation detection techniques or by analyzing different strains/viruses of this family. Systems involving other hosts, such as the *Cafeteria roenbergensis*-croV-mavirus trio[19,27,68], might depend on DNA methylation to regulate their intricate interactions. DNA methylation is also a key factor in the switch between latent and integrated forms of some viruses[12]. In the *Cafeteria roenbergensis*-croV-mavirus context, one might wonder about the role DNA methylation could play in the maintenance of the host integrated mavirus provirophage, or in its awakening in the presence of croV infections.

R-M systems provide the carrying bacteria an immediate protection against the most lethal bacteriophages present it its environment[6]. Our work suggests that DNA methylation is equally important in giant viruses and involved in several types of interactions depending on the presence or absence of REase activity, and on the strictly cytoplasmic or nucleus dependency of their replication cycle. In chloroviruses, R-M systems offer a way to attack host DNA and exploit its nucleotide pool[14]. Our work on marseilleviruses now suggests that they act as an offensive weapon against competing pathogens. By contrast, the many orphan MTases found in giant viruses are potential self-defense weapons against other pathogens bearing active R-M systems with similar targets. Therefore, DNA methylation might allow giant viruses to face the fierce battles taking place in their amoebal hosts. In that context, it seems odd that the most frequent giant viruses in the environment that we studied, the *Mimiviridae* members, are apparently devoid of DNA methylation. However, one might remember that bacteria developed different lethal weapons to survive phage invasion: R-M and CRISPR systems. The many other REases found in the giant viruses' genomes could thus be involved in other competition/resistance processes such as the Mimivire system suggested to be directed against the parasitic virophage competing with some *Mimiviridae* members for its replication in *Acanthamoeba*[70]. Alongside toxin-antitoxin systems[69,71], Mimivire or other yet to be discovered CRISPR-like systems, DNA methylation might be part of the giant viruses' arsenal to cope with their numerous competitors.

## Methods

**Cedratvirus kamchatka characterization.** Cedratvirus kamchatka was isolated from muddy grit soil collected near a lake at kizimen volcano, Kamchatka (Russian Federation N 55° 05′ 50 E 160° 20′ 58). The sample was resuspended in phosphate buffer saline containing ampicillin (100 μg mL$^{-1}$), chloramphrenicol (30 μg mL$^{-1}$) and kanamycin (25 μg mL$^{-1}$), and an aliquot was incubated with *A. castellanii* Neff (ATCC 30010$^{TM}$) cells (2000 cells per cm$^2$) adapted to 2.5 μg mL$^{-1}$ of Amphotericin B (Fungizone), in protease-peptone-yeast-extract-glucose (PPYG) medium. Cultures exhibiting infectious phenotypes were recovered, centrifuged 5 min at $500 \times g$ to eliminate the cells debris and the supernatant was centrifuged for 1 h at $16,000 \times g$ at room temperature. T75 flasks were seeded with 60,000 cells per cm$^2$ and infected with the resuspended viral pellet. After a succession of passages, viral particles produced in sufficient quantity were recovered and purified. The viral

pellet was resuspended in PBS and loaded on a 1.2 to 1.5 density cesium gradient. After 16 h of centrifugation at 200,000 × g, the viral disk was washed three times in PBS and stored at 4 °C.

The genomic DNA of cedratvirus kamchatka was recovered from $2 \times 10^{10}$ purified particles resuspended in 300 µL of water incubated with 500 µL of a buffer containing 100 mM Tris-HCl pH 8, 1.4 M NaCl, 20 mM Na2EDTA, 2% (w/v) CTAB (cetyltrimethylammonium bromide), 6 mM DTT and 1 mg mL⁻¹ proteinase K at 65 °C for 90 min. After treatment with 0.5 mg mL⁻¹ RNase A for 10 min, 500 µL of chloroform was added and the sample was centrifuged at 16,000 × g for 10 min at 4 °C. One volume of chloroform was added to the supernatant and centrifuged at 16,000 × g for 5 min at 4 °C. The aqueous phase was incubated with 2 volumes of precipitation buffer (5 g L⁻¹ CTAB, 40 mM NaCl, pH 8) for 1 h at room temperature and centrifuged for 5 min at 16,000 × g. The pellet was then resuspended in 350 µL of 1.2 M NaCl and 350 µL of chloroform was added and centrifuged at 16,000 × g for 10 min at 4 °C. The aqueous phase was mixed to 0.6 volume of isopropanol, centrifuged for 10 min at room temperature and the pellet was washed with 500 µL 70% ethanol, centrifuged again and resuspended with nuclease-free water.

The DNA was sequenced using the PacBio SMRT technology, resulting in 868 Mb of sequence data (76,825 reads). SMRT reads were filtered using the SMRTanalysis package version 2.3.0. We then used Flye[72] version 2.4.2 to perform the de novo genome assembly with the pacbio_raw and $g = 450,000$ parameters. The assembly resulted in two distinct contigs (one of 466,767 nt and a second of 6049 nt). A pulsed field gel electrophoresis (PFGE) confirmed the genome size and its circular structure (Supplementary Fig. 4A). The smaller contig, not seen on the PFGE, potentially corresponds to an assembly artifact. Finally the assembly was subsequently polished using the Quiver tool from SMRTanalysis.

The cedratvirus kamchatka gene annotation was performed using GeneMarkS[73] with the virus option. Only genes predicted to encode proteins of at least 50 amino acids were kept for functional annotation. These proteins were aligned against the NR and Swissprot databases using BlastP[74] (with an E-value cutoff of 10⁻⁵) and submitted to CD search[39], to InterProScan[75] with the Pfam, PANTHER, TIGRFAM, SMART, ProDom, ProSiteProfiles, ProSitePatterns and Hamapts databases, and to Phobius[76]. The genome was then manually curated according to these data.

**SMRT resequencing.** Pithovirus sibericum, mollivirus sibericum and megavirus vitis/zamilon vitis/megavirus vitis transpoviron genomic DNAs were extracted from $2 \times 10^{10}$ purified particles using the PureLink Genomic DNA Extraction Mini Kit (Thermo Scientific) according to the manufacturer protocol. For pithovirus sibericum, we performed two successive purifications, and added 10 mM DTT in the lysis buffer for the first one.

The sequencing of pithovirus sibericum, mollivirus sibericum and megavirus vitis/zamilon vitis/megavirus vitis transpoviron performed using PacBio SMRT technology resulted in, respectively, 779 Mb (143,675 reads), 371 Mb (66,453 reads), and 997 Mb (66,464 reads) of sequence data. A second SMRT sequencing was performed on pithovirus sibericum and mollivirus sibericum viral DNA after WGA amplification using the Illustra GenomiPhi V2 DNA Amplification kit (GE Healthcare) according to the manufacturer instructions. This resulted in 632 Mb (78,685 reads) and 346 Mb (55,797 reads) of sequences for pithovirus sibericum and mollivirus sibericum WGA amplified DNA.

**Identification of methylated motifs.** Methylated motifs were identified using the modification and motif analysis module from the SMRTanalysis package using the datasets described in Supplementary Table 1 and the corresponding reference genome sequences. In addition we used the per-base resolution file of IPD ratios calculated by SMRTanalysis to compute the global IPDr of detected motifs and predicted MTases targets.

**Annotation of MTases, REases, and target prediction.** We analyzed the MTases and REases encoded in giant viruses genomes based on published annotations when available, as well as a combination of CD search and Interproscan protein domain search tools analyses[39,40]. In addition, we performed blast alignments against the REbase database[41] to support the annotations and to predict their targets when possible.

**Phylogenies.** All the phylogenies were calculated from the protein multiple alignments of homologous sequences computed using Expresso[77], Mafft[78], Mcoffee[79], and Clustal Omega[80]. We next selected the best multiple alignment using TrimAI[81].

For phylogenies based on single genes we identified homologs using BlastP against the NR database with an E-value cutoff of 10⁻¹⁰. IQtree[82] was used to compute the tree using the best model as defined by ModelFinder[83]. Bootstrap values were computed using the UFBoot method[84].

Phylogenies of viral families (Fig. 2 and Supplementary Fig. 9) were based on the multiple alignments of strictly conserved single copy genes as defined by the OrthoFinder algorithm[85]. Next we used IQtree[82] with the -p option to compute the tree using the best model of each partitioned alignment. Each orthogroup (i.e., cluster of single copy orthologues) and the corresponding best models found by IQtree[83,86] are listed in Supplementary Data 1.

**Selection pressure measurements.** We performed codon-based multiple alignments of each subgroup of giant viruses MTases (Fig. 5) using protein multiple alignments (see Phylogenies Methods) and nucleotide sequences. The ω were computed for each gene of interest using Codeml[46] through the ETE framework[87]. The M0 model, which considers a unique ω for the whole tree, was first computed. Genes that were too divergent, i.e., where multiple substitutions might have occurred (dS > 1.5), were excluded. We then calculated the ω of the remaining genes using the B_free model, which assigns two distinct ω (one for the branch of interest and one for the rest of the tree), and the B_neut model with a fixed $ω = 1$ for the gene of interest. LTR tests with a p-value cutoff of 0.05 were then performed to select the best model and to decide whether ω were significantly different from one.

**DNA restriction experiments.** Mollivirus sibericum, melbournevirus and noumeavirus genomic DNA were extracted using the PureLink Genomic DNA Extraction Mini Kit (Thermo Scientific) according to the manufacturer protocol. A. castellanii genomic DNA was extracted using the Wizard® Genomic DNA Purification Kit (Promega). The DNA were digested with 10 units of the appropriate restriction enzymes (New England Biolabs) for 1 h at 37 °C and loaded on a 1% agarose gel.

**Mollivirus sibericum-infected A. castellanii cells DNA extraction.** A. castellanii overexpressing the ml_216-GFP fusion and wild-type cells were grown in T25 flasks. They were infected with mollivirus sibericum at MOI 100. After 1 h of infection at 32 °C, cells were washed three times with PPYG to eliminate the excess of viruses. For each infection time (1 h to 6 h), a T25 flask was recovered and cells were centrifuged for 5 min at 1000 × g. DNA was extracted using the Wizard® Genomic DNA Purification Kit (Promega) according to the manufacturer's protocol and loaded on a 0.8% agarose gel.

**Melbournevirus R-M system MTase and REase expression timing.** A. castellanii cells were grown in T25 flasks and infected with melbournevirus at MOI 50. After 15 min of infection at 32 °C, cells were washed 3 times with PPYG to eliminate the excess of viruses. For each infection time (15 min to 5 h), a T25 flask was recovered and cells were centrifuged for 5 min at 1000 × g. RNA was extracted using the RNeasy Mini kit (QIAGEN) according to the manufacturer's protocol. Briefly, cells were resuspended in the provided buffer and disrupted by −80 °C freezing and thawing, and shaken vigorously. Total RNA was eluted with 50 µL of RNase free water. Total RNA was quantified on the nanodrop spectrophotometer (Thermo Scientific). Poly(A) enrichment was performed (Life Technologies, Dynabeads oligodT₂₅) and first-strand complementary DNA (cDNA) poly(A) synthesis was performed with the Smart-Scribe Reverse Transcriptase (Clontech Laboratories) using an oligo(dT)₂₄ primer and then treated with RNase H (New England Biolabs). For each time point, PCR reactions were performed using mel_015 and mel_016 genes specific primers and one unit of Phusion DNA Polymerase (Thermo Scientific) in a 50 µL final volume.

**Motif enrichment in intergenic regions.** Analysis of motif enrichment in intergenic regions was done by computing the number of motifs identified in these regions and in coding regions. We next shuffled coding coordinates 1000 times and calculated the same values. Z-scores were calculated from these randomizations and transformed in empirical p-values. We excluded the CCTNAGG motifs from this calculation as there were not enough occurrences of this motif (Fig. 1) to compute accurate p-values.

**Reporting summary.** Further information on research design is available in the Nature Research Reporting Summary linked to this article.

## Data availability

Data supporting the findings of this work are available within the paper and its Supplementary Information files. The raw SMRT sequence datasets generated and analyzed in the current study were deposited in the Sequence Read Archive database under the following accession PRJNA612691. In addition individual datasets accessions are all reported in the Supplementary Table 1. The assembled cedratvirus kamchatka genome has been deposited to the GenBank database under the following accession MN873693. MTases annotations were performed using the REbase database [http://rebase.neb.com/rebase/rebase.html] and the cedratvirus kamchatka gene annotations using the Blast NCBI NR (GenBank CDS translations+PDB + SwissProt+PIR + PRF) database and the Uniprot-Swissprot [https://www.uniprot.org/uniprot/] database. The source data underlying Fig. 1, Supplementary Figs. 1 and 10, Supplementary Tables 3–5 are provided as source data file.

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

## Acknowledgements

We are deeply indebted to our volunteer collaborator Alexander Morawitz for collecting the Kamchatka soil samples. We thank the PACA Bioinfo platform for computing support. This project has received funding from the European Research Council (ERC) under the European Union's Horizon 2020 research and innovation program (grant agreement No 832601), from the FRM prize Lucien Tartois and from CNRS (PRC1484-2018) to C. Abergel. The funding bodies had no role in the design of the study, analysis, and interpretation of data and in writing the manuscript.

## Author contributions

S.J., C.A. and M.L. designed research. S.J. performed most of the experiments. S.R. assembled, annotated, and analyzed the cedratvirus kamchatka genome. J.M.A. isolated cedratvirus kamchatka and provided research assistance. S.J., C.A. and J.M.C. contributed to manuscript writing. M.L. directed the research, carried most of the data analysis, and wrote the paper.

## Competing interests

The authors declare no competing interests.
