## [Peer Review File · Nature Communications]

Reviewers' Comments:

Reviewer #1:

Remarks to the Author:

Judy et al provide an examination of DNA methylation patterns in giant viruses. The authors examine public data and generate SMRT data of their own to assess the level of DNA methylation in giant virus genomes from diverse viral families. The authors also provide bioinformatic analysis of predicted methyltransferases and endonucleases (so-called R-M systems) to identify the enzymes putatively responsible. They also provide an evolutionary analysis of the enzymes.

Overall this is a rigorous study that provides important insights into methylation of giant viruses. This is an important advance since methylation likely plays a role in host-virus interaction and viral coinfection dynamics. I am not an expert on interpreting methylation patterns from SMRT data, so I cannot evaluate the methods used for that in detail, but based on my limited knowledge and the information provided here I find the results convincing. I have a few specific comments below:

-Figure S8 is important and should be included in the main text, in my opinion. The evolutionary history of MTases is quite fascinating and an important aspect of this study, so highlighting this in the main text is warranted. I would also include a legend with the colors explained so the reader can better interpret the findings. Lastly, I find it difficult to examine the bootstrap support when it is provided as a color range- I think it would be preferable to include solid circles on the nodes that have UFBoot support > 80. Also, how was this tree rooted, and using what rationale?

-The authors observe that Pithovirus sibericum encodes two methyltransferases that are expressed, yet it's genome appears unmethylated. They conclude that "P. sibericum MTase-like proteins do not methylate the viral DNA even though they are expressed". Is it possible that these MTases methylate the genome in different infection conditions? I agree the result is puzzling and certainly does not provide evidence for methylation, but there may be some mechanism for inhibiting these enzymes that could explain this result. In other conditions it remains possible that some methylation could take place. The authors make similar claims about the MTases of two Mimiviridae, but once again I am not sure that conclusions about the activity of specific enzymes is warranted with the data presented here. The authors may wish to restate these conclusions to allow for some ambiguity- it may be that future studies find cryptic methylation patterns that occur in specific circumstances.

-It would be useful to have percent methylation statistics in the abstract (max number of motifs, detected, etc).

-"Transpoviron" should be defined in the introduction- this term is used often in this manuscript but many readers may not have encountered it before.

-There are no line numbers, at least in the document I downloaded, which makes it a bit difficult to provide detailed comments. Please provide those in the future.

-I find it interesting that the predicted methylation site agreed with that detected in 78% of cases. This suggests methylation site predictions based on protein homology are highly reliable. This could be emphasized more in the Discussion.

-For the dN/dS ratios, were values with high dS filtered out? Generally this is done to remove divergent comparisons in which multiple substitutions have occurred. Generally dS cutoffs of 1 or 2 are used. More detail on the methods here would be useful.

-More details should be provided regarding the single-copy genes used for viral phylogenies. It is unclear to me what genes were used, so a table or supplementary spreadsheet should be provided. Also, for IQTREE, please provide the best-fit model that was ultimately used. If the ModelFinder utility was used, please cite that paper: <https://doi.org/10.1038/nmeth.4285>

Some wording suggestions:

-Page 9: "comfort this observation" -> "provide further evidence to test this hypothesis"

-Viral family names are sometimes not italicized.

-Page 2 "all kinds of environments" -> "many/diverse environments"

-Abstract- "if some viral MTases undergoe pseudogenization" -> "although some viral MTases appear to undergo pseudogenization".

Abstract: "different kind" -> "diverse forms"

Frank Aylward

Reviewer #2:

Remarks to the Author:

The manuscript provides a very good investigation into the methylation systems of giant viruses. In the first line of the abstract, the first line of the introduction and the first paragraph the authors nominate "epigenetic", but no results is provided and or more specific discussion is attempted. It should not be forgotten that DNA methylation also if only responsible for restriction resistance, is still an epigenetic modification. It could also be of interest to look in the final paragraph of the discussion on alternative epigenetic modification systems (see for example <https://doi.org/10.1093/nar/gkt573>). As the discussion is very long, maybe it could be shortened and to generate some space to raise some hypotheses for an epigenetic role of the described modifications.

Specific comments

At the start of the results authors describe that they "gathered data from previously published genomes". It would be very informative to define in the results text which had been sequenced, deposited and described by your group, as this is absolutely not evident in the text. As it stands your contribution to the field is not evident. I was afraid that the genomes were all from a different lab which could have been potentially working on the same paper.

Table S1 check

Authors state that "In most cases (78%) we found an agreement between the predicted targets and the detected methylated motifs". This 78% value is not evident from figure 1, please provide detail in the text.

In Fig 2 it is difficult to understand which genomes were sequenced in this work and which were from other work, as I was tempted to look up for the virus name of Figure 1 and only one is clearly evident. Please make this clear in the legend.

Check Fig S2

The statement "Thus, Melbournevirus protects its own genome from its encoded R-M system REase" should be revised as the work demonstrate that the methylase methylates the GATC target, but authors do not demonstrate that the restriction enzyme cleaves unmethylated GATC – you only infer this. An experiment using a Melbournevirus extract which digests unmethylated DNA but not methylated DNA would show this.

FigS8

Authors refer to a Mollivirus MTase homologue is found in *A. polyphaga*. It would be of interest to have a short look at the surrounding genes on the *Acanthamoeba* genome to check if there is evidence

of other virus derived inserted genes.

Reviewer #3:

Remarks to the Author:

Giant viruses have been identified in the past decade or so, and are fascinating biological objects. The authors are recognized members of the field. Here, they report of the DNA modification machinery in giant viruses, mostly by re-analyzing available SMRT-seq data to detect indications of methylated bases. They also perform searches to identify putative DNA methyltransferases and their related Restriction Enzymes. Finally, a few predictions are validated by digesting viral DNA with methylation-sensitive restriction enzymes.

The paper is well written. The experiments are well presented and generally convincing. That being said, I don't feel I can offer a strong recommendation for publication in Nat Comm, for 2 reasons.

1-I am not sure the novelty and general interest are high enough. On the one hand, to the best of my knowledge, nothing is known about DNA methylation in giant viruses. But then again, as the authors point out, other viruses of eukaryotes have DNA methyltransferases and methylated DNA (ascoviridae, iridoviridae). And the evolution of giant virus in relation to their hosts has been discussed in more depth in other papers, eg Guglielmini PNAS 2019 PMID:31506349; Schulz Nature 2020 PMID:31968354.

2-There is no functional data. At present the only indication that these modifications could do something important is the fact that they undergo selective pressure. But there are no experiments to back this up. Of course, I realize they may be hard to do within this system, but the fact remains that there is no mechanistic data at all.

Finally, it would have been nice to show that the modifications exist by additional techniques besides SMRT-seq and restriction digest, for instance immunoprecipitation of m6A. But this is not my main concern.

Reviewer #1 (Remarks to the Author):

Jeudy et al provide an examination of DNA methylation patterns in giant viruses. The authors examine public data and generate SMRT data of their own to assess the level of DNA methylation in giant virus genomes from diverse viral families. The authors also provide bioinformatic analysis of predicted methyltransferases and endonucleases (so-called R-M systems) to identify the enzymes putatively responsible. They also provide an evolutionary analysis of the enzymes.

Overall this is a rigorous study that provides important insights into methylation of giant viruses. This is an important advance since methylation likely plays a role in host-virus interaction and viral coinfection dynamics. I am not an expert on interpreting methylation patterns from SMRT data, so I cannot evaluate the methods used for that in detail, but based on my limited knowledge and the information provided here I find the results convincing. I have a few specific comments below:

R1.1

-Figure S8 is important and should be included in the main text, in my opinion. The evolutionary history of MTases is quite fascinating and an important aspect of this study, so highlighting this in the main text is warranted. I would also include a legend with the colors explained so the reader can better interpret the findings. Lastly, I find it difficult to examine the bootstrap support when it is provided as a color range- I think it would be preferable to include solid circles on the nodes that have UFBoot support > 80. Also, how was this tree rooted, and using what rationale?

As requested by the reviewer Fig. S8 is now in the main text (new Fig. 5) with the corresponding extended legend. We also changed the bootstrap labels using solid circles on branches with branch support values >80. Finally the tree was rooted using the midpoint rooting technique. It is now mentioned in the legend.

The new Fig 5 legend is the following:

Phylogenetic tree of the giant viruses' MTases along with prokaryotic and eukaryotic homologues. The blue triangles mark viral genes, the red ones eukaryotic genes and the unmarked genes are prokaryotic. The tree was computed using the LG+R6 model from a multiple alignment of 678 informative sites. Bootstrap values were computed using the UFBoot (113) method from IQtree (111). All branches with support value > 80 are highlighted using purple circles. The GenBank accessions and taxonomic assignments extracted from GenBank entries are shown. The tree was rooted using the midpoint rooting method. The tree was split into five subgroups highlighted using different colors (blue, orange, red, purple and green).

R1.2

-The authors observe that Pithovirus sibericum encodes two methyltransferases that are expressed, yet its genome appears unmethylated. They conclude that "P. sibericum MTase-like proteins do not methylate the viral DNA even though they are expressed". Is it possible that these MTases methylate the genome in different infection conditions? I agree the result is puzzling and certainly does not provide evidence for methylation, but there may be some mechanism for inhibiting these enzymes that could explain this result. In other conditions it remains possible that some methylation could take place. The authors make similar claims about the MTases of two Mimiviridae, but once again I am not sure that conclusions about the activity of specific enzymes is warranted with the data presented here. The authors may wish to restate

these conclusions to allow for some ambiguity- it may be that future studies find cryptic methylation patterns that occur in specific circumstances.

We modified the discussion paragraph on *P. sibericum* lack of DNA methylation to take these remarks into account (see p9 lines 337-341). More specifically, beyond the fact that SMRT-seq might not be sensitive enough to detect cryptic methylation, we also propose that viral DNA might be transiently methylated and that methylation marks might be erased. We also soften the claims of the Mimiviridae lack of methylation as suggested by the reviewer in another paragraph of the discussion (p12 lines 442-444).

The modified sentences regarding the pithovirus MTase are the following:

Here, uncharacterized mutations in a critical part of the protein might be at play and explain the loss of function. Alternatively, although less likely, the lack of methylation could be the result of transient methylation where methylation marks of viral DNA are eventually erased by an unknown N6-methyl adenine demethylase. Finally, one cannot exclude that SMRT-seq is not sensitive enough to detect cryptic methylation of pithovirus DNA.

And for the Mimiviridae

However future studies might reveal DNA methylation of the *Mimiviridae*-viroplasm system using different methylation detection techniques or by analyzing different strains/viruses of this family.

R1.3

-It would be useful to have percent methylation statistics in the abstract (max number of motifs, detected, etc).

We modified the abstract to incorporate methylation statistics in the following way:

Here we show that DNA methylation is widespread in giant viruses, affecting 2/3 of the tested families, although unevenly distributed with 35-fold difference in methylation sites relative frequencies.

R1.4

-“Transpoviron” should be defined in the introduction- this term is used often in this manuscript but many readers may not have encountered it before.

We now define “transpoviron” in the introduction along with the appropriate references (see p2 lines 56-58).

Intriguingly some *Mimiviridae* members are infected by smaller 20-kb-dsDNA viruses, dubbed “virophages” (38–40) and sometimes found in association with 7-kb-DNA episomes called transpovirons (41, 42).

R1.5

-There are no line numbers, at least in the document I downloaded, which makes it a bit difficult to provide detailed comments. Please provide those in the future.

Done

R1.6

-I find it interesting that the predicted methylation site agreed with that detected in 78% of cases. This suggests methylation site predictions based on protein homology are highly reliable. This could be emphasized more in the Discussion.

We emphasized this point as suggested by the reviewer but kept it in the result section (see p4 lines 112-115). As pointed out by the Reviewer 2 the discussion is rather long and we had to incorporate other points in this section.

In 15 out of the 19 cases (79%) where an MTase target could be predicted or a motif detected, we found an agreement between the two (Fig. 1). It is worth mentioning that this result also highlights the reliability of MTases' targets predictions based on protein homology.

R1.7

-For the dN/dS ratios, were values with high dS filtered out? Generally this is done to remove divergent comparisons in which multiple substitutions have occurred. Generally dS cutoffs of 1 or 2 are used. More detail on the methods here would be useful.

Thanks to the reviewer for raising this point. We now apply a filter to exclude divergent sequences in dN/dS calculations. Only sequences with $dS < 1.5$ were kept. This slightly lowered the number of genes for which a dN/dS ratio can be calculated (20 instead of 24) but the conclusions remain identical (see revised table S4). This filtering is now mentioned in the Methods section.

R1.8

-More details should be provided regarding the single-copy genes used for viral phylogenies. It is unclear to me what genes were used, so a table or supplementary spreadsheet should be provided. Also, for IQTREE, please provide the best-fit model that was ultimately used. If the ModelFinder utility was used, please cite that paper: <https://doi.org/10.1038/nmeth.4285>

Again thanks to the reviewer for his suggestions. The Methods section related to viral families has been changed to incorporate these comments. We now provide a supplemental file (Supplementary data 1) with the groups of orthologous genes used to compute the Marseilleviridae (Fig 2) and Pandoraviridae phylogenies (now Fig. S9) as well as the best-fit evolutionary model identified by ModelFinder. The references to ModelFinder and to the partitioned phylogeny procedure are now duly mentioned.

R1.10

Some wording suggestions:

-Page 9: "comfort this observation" -> "provide further evidence to test this hypothesis"

Corrected

-Viral family names are sometimes not italicized.

Corrected

-Page 2 "all kinds of environments" -> "many/diverse environments"

Corrected

-Abstract- “if some viral MTases undergo pseudogenization” -> “although some viral MTases appear to undergo pseudogenization”.

Corrected

Abstract: “different kind” -> “diverse forms”

Corrected

Reviewer #2 (Remarks to the Author):

R2.1

The manuscript provides a very good investigation into the methylation systems of giant viruses.

In the first line of the abstract, the first line of the introduction and the first paragraph the authors nominate “epigenetic”, but no results is provided and or more specific discussion is attempted. It should not be forgotten that DNA methylation also if only responsible for restriction resistance, is still an epigenetic modification. It could also be of interest to look in the final paragraph of the discussion on alternative epigenetic modification systems (see for example <https://doi.org/10.1093/nar/gkt573>). As the discussion is very long, maybe it could be shortened and to generate some space to raise some hypotheses for an epigenetic role of the described modifications.

We thank the referee for pointing out this reference that allowed us to elaborate on the epigenetic role of DNA methylation in giant viruses besides restriction and restriction-resistance functions. We are still in favor of a restriction related role of MTases in giant viruses for the following reasons: i) methylated sites are uniformly distributed on the genomes, not clustered in specific regions, ii) all sites are methylated so not in favor of being targeted by regulators and thus protected against modification by the MTase, iii) the sites are not specifically enriched in promoter/intergenic regions that would point to a gene regulatory role and iv) it fits better with the phylogenetic pattern of giant viruses MTases and their apparent patchy distribution. However we modified the discussion to incorporate other epigenetic aspects and cited this reference and others.

See p11 line 416-to page 12 line 430:

Besides *Marseilleviridae* family members, giant viruses for which we identified MTases and observed DNA methylation do not seem to encode cognate REases. Such so-called orphan MTases are common in bacteria where they regulate various biological processes, such as replication initiation, mismatch repair or gene expression (7–11). Accordingly, the targeted genomic positions are not uniformly distributed, with hotspots and coldspots of fully-, hemi- and unmethylated sites. Likewise, DNA modifications in phages have epigenetic roles beyond R-M systems. For instance the P1 phage Dmt-encoded MTase is involved in the control of DNA concatemers cleavage at the initiation of DNA packaging (91–93). Again the methylated sites are clustered in the so-called pac regions. One could hypothesize a similar replication-related epigenetic role of orphan giant viruses MTases. However, giant viruses’ genomes exhibit unimodal distributions of IPDr values and the corresponding motifs are globally uniformly distributed (Fig. S11). In the case of bacterial orphan MTases involved in gene regulations the methylated sites tend to be located in the upstream non-coding regions of the regulated genes (48). This is not true

for giant viruses where methylated motifs are not enriched in intergenic regions (p-values > 0.05, see Methods) and thus not favoring such an epigenetic role.

R2.2

At the start of the results authors describe that they “gathered data from previously published genomes”. It would be very informative to define in the results text which had been sequenced, deposited and described by your group, as this is absolutely not evident in the text. As it stands your contribution to the field is not evident. I was afraid that the genomes were all from a different lab which could have been potentially working on the same paper.

Table S1 check

All the SMRT-seq datasets used in this study were produced by our group either from previously published genomes or specifically for this work (see revised Table S1). We modified the text and added corresponding references in the results text to avoid ambiguities (see p4 lines 90-99).

We gathered PacBio SMRT data of diverse families from previously published genomes sequenced by our group to analyze the DNA methylation profile of a wide range of giant viruses. SMRT genomic data were collected for the following viruses: the *Mimiviridae* member mousmouvirus australiensis and its associated transpoviron (41), the *Marseilleviridae* member melbournevirus (51) and five pandoraviruses (pandoravirus celtis (52), pandoravirus dulcis (43), pandoravirus neocaledonia (43), pandoravirus quercus (43) and pandoravirus salinus (43)). In addition, we resequenced on the PacBio platform the complete genomes of mollivirus sibericum, pithovirus sibericum, the *Lavidaviridae* member zamilon vitis and the megavirus vitis *Mimiviridae* member together with its associated transpoviron. Finally we sequenced a newly isolated strain of cedratvirus (cedratvirus kamchatka). The used datasets are listed in Supplementary Table 1.

R2.3

Authors state that “In most cases (78%) we found an agreement between the predicted targets and the detected methylated motifs”. This 78% value is not evident from figure 1, please provide detail in the text.

This percentage (rounded to 79%) corresponds to 15 cases where predicted targets matched identified motifs (filled red and blue circles in Fig. 1) out of the 19 cases where either a target was predicted or a methylated motif was detected (at least a red and/or blue filled circle). This is now detailed in the results (p4 lines 112-115) in the following way:

In 15 out of the 19 cases (79%) where an MTase target could be predicted or a motif detected, we found an agreement between the two (Fig. 1). It is worth mentioning that this result also highlights the reliability of MTases’ targets predictions based on protein homology.

R2.4

In Fig 2 it is difficult to understand which genomes were sequenced in this work and which were from other work, as I was tempted to look up for the virus name of Figure 1 and only one is clearly evident. Please make this clear in the legend.

The phylogeny presented in this figure was performed on the published *Marseilleviridae* genomes. Among them three were sequenced by our group (Melbournevirus, Noumeavirus and Port-Miou virus) and one of them (Melbournevirus) was sequenced using SMRT sequencing. The raw data from the latter was used in

this study to explore its methylome. To clarify this we deleted the Table S1B and moved the GenBank IDs to the legend of the figure alongside with the references of the corresponding publications.

R2.5

The statement “Thus, Melbournevirus protects its own genome from its encoded R-M system REase” should be revised as the work demonstrate that the methylase methylates the GATC target, but authors do not demonstrate that the restriction enzyme cleaves unmethylated GATC – you only infer this. An experiment using a Melbournevirus extract which digests unmethylated DNA but not methylated DNA would show this.

As mentioned in the manuscript (p6 lines 174-176) proteomic data of the Melbournevirus particle show that the MEL_015 REase is not packaged in the virion, only the MTase is. Thus the experiment suggested by the referee to use Melbournevirus particle extract to digest unmethylated DNA cannot be done. Consequently, we softened our claims on the presence of a functional R-M system in the manuscript to take this remark into account. Specifically we modified the text (p5 lines 152-153) in the following way:

Thus, assuming that MEL_015 REase is functional, we can infer that melbournevirus is able to protect its own genome from its encoded R-M system digestion.

R2.6

FigS8

Authors refer to a Mollivirus MTase homologue is found in *A. polyphaga*. It would be of interest to have a short look at the surrounding genes on the *Acanthamoeba* genome to check if there is evidence of other virus derived inserted genes.

This homolog is encoded in a short *A. polyphaga* Linc contig (LQHA01005312.1) without further evidence of virus-derived inserted genes.

Reviewer #3 (Remarks to the Author):

Giant viruses have been identified in the past decade or so, and are fascinating biological objects. The authors are recognized members of the field. Here, they report of the DNA modification machinery in giant viruses, mostly by re-analyzing available SMRT-seq data to detect indications of methylated bases. They also perform searches to identify putative DNA methyltransferases and their related Restriction Enzymes. Finally, a few predictions are validated by digesting viral DNA with methylation-sensitive restriction enzymes.

R3.1

The paper is well written. The experiments are well presented and generally convincing. That being said, I don't feel I can offer a strong recommendation for publication in Nat Comm, for 2 reasons.

We thank the referee for pointing out the quality of this work and try to respond to his main concerns.

R3.2

1-I am not sure the novelty and general interest are high enough. On the one hand, to the best of my knowledge, nothing is known about DNA methylation in giant viruses. But then again, as the authors point out, other viruses of eukaryotes have DNA methyltransferases and methylated DNA (ascoviridae, iridoviridae). And the evolution of giant virus in relation to their hosts has been discussed in more depth

in other papers, eg Guglielmini PNAS 2019 PubMed ID 31506349; Schulz Nature 2020 PubMed ID 31968354.

The work of Guglielmini et al. addresses an important question, namely the deep evolutionary relationships of giant viruses and more broadly NCLDVs. Although these relationships are far from being fully understood, this is not the purpose of the present work. Our point was not to tackle the global evolution of giant viruses but to show the intricate (and “fascinating” as pointed out by the reviewer 1, see R1.1) evolutionary history of the MTases within these viruses and other organisms with whom they interact (other viruses, bacteria and host). To our knowledge this has not been done before and this is not addressed by the two cited references.

The work of Guglielmini et al. and Schulz et al establish a shared evolutionary history between giant viruses based on a handful of shared genes, actually only 3 genes (out of thousands encoded by some giant viruses) when taking into account the viral families explored in this work. But even if they do share a putative very ancient common ancestor, their work (among others) highlights the great divergence between the families in terms of genome sequence and structure, gene content, encoded molecular functions, infection strategy, hosts and physiology. Thus, in our opinion the fact that DNA methylation has been described in a couple of papers in Adenoviruses and Iridoviruses does not preclude to explore this function in diverse families. On the contrary we believe that it urges to challenge how broadly and important DNA methylation impacts giant viruses.

The work of Schulz et al, exclusively based on metagenomic data, shows the enormous diversity of giant viruses in diverse environments. In our opinion this highlights the need to understand their physiology and identify the molecular functions that drive the interactions between them and with their hosts. According to our work DNA methylation is one of them. We believe that our work, based on focused omics data and experimental data, complements the bigger picture provided by Schultz et al and other papers on this topic.

R3.3

2-There is no functional data. At present the only indication that these modifications could do something important is the fact that they undergo selective pressure. But there are no experiments to back this up. Of course, I realize they may be hard to do within this system, but the fact remains that there is no mechanistic data at all.

Finally, it would have been nice to show that the modifications exist by additional techniques besides SMRT-seq and restriction digest, for instance immunoprecipitation of m6A. But this is not my main concern.

In addition to the identification of global DNA methylation using SMRT-seq we also performed restriction experiments that revealed a protective role against DNA degradation by REases in Marseilleviridae. Therefore we believe that we partially addressed the molecular function of some MTases. Unfortunately, there are no molecular tools available that would allow to go deeper in the mechanism of DNA methylation in these viruses. Ideally one would think of experiments quantifying the impact of methylation on the viruses' fitness. This would require to genetically modify the viruses to knock out the MTases and test it on a population using single cell infections experiments. Currently we do not have the experimental setup and genetic tools to perform such experiments on these systems, but we wish to have such tools in the future.

Reviewers' Comments:

Reviewer #1:

Remarks to the Author:

The authors have addressed my comments in their revised manuscript, and I have no further concerns.

Reviewer #2:

Remarks to the Author:

authors have addressed all issues raised during the review

Reviewer #1 (Remarks to the Author):

The authors have addressed my comments in their revised manuscript, and I have no further concerns.

Reviewer #2 (Remarks to the Author):

Authors have addressed all issues raised during the review.

We thank the referees for their comments and for the time they spent on our manuscript.